# Osteopontin (OPN/SPP1), a Mediator of Tumor Progression, Is Regulated by the Mesenchymal Transcription Factor Slug/SNAI2 in Colorectal Cancer (CRC)

**DOI:** 10.3390/cells11111808

**Published:** 2022-05-31

**Authors:** Katyana Amilca-Seba, Tuan Zea Tan, Jean-Paul Thiery, Lila Louadj, Sandrine Thouroude, Anaïs Bouygues, Michèle Sabbah, Annette K. Larsen, Jérôme A. Denis

**Affiliations:** 1Cancer Biology and Therapeutics, Centre de Recherche Saint-Antoine (CRSA), 75571 Paris, France; katyana.amilca@inserm.fr (K.A.-S.); lila.louadj@sorbonne-universite.fr (L.L.); sandrine.thouroude@inserm.fr (S.T.); anais.bouygues@inserm.fr (A.B.); michele.sabbah@inserm.fr (M.S.); annettelarsen004@gmail.com (A.K.L.); 2Institut National de la Santé et de la Recherche Médicale (INSERM) U938, 75012 Paris, France; 3Institut Universitaire de Cancérologie (IUC), Sorbonne Université, 75005 Paris, France; 4Center for Translational Medicine, Cancer Science Institute of Singapore, National University of Singapore, Singapore 117599, Singapore; csittz@nus.edu.sg; 5Guangzhou Laboratory, Guangzhou 510530, China; tjp@grmh-gdl.cn; 6Centre National de la Recherche Scientifique (CNRS), 75016 Paris, France; 7Department of Endocrinology and Oncology Biochemistry, Pitié-Salpetrière Hospital, 075013 Paris, France

**Keywords:** colorectal cancer (CRC), epithelial–mesenchymal transition (EMT), Slug/*SNAI2* transcription factor, osteopontin (OPN/SPP1)

## Abstract

**Simple Summary:**

Expression of the transcription factor Slug/SNAI2 is associated with the epithelial–mesenchymal transition (EMT) and is correlated with poorer disease-free survival in colorectal cancer (CRC). In order to decipher the basis for the Slug-mediated aggressive phenotype, we conducted RNAseq experiments with a panel of HT-29 CRC cells expressing different levels of Slug, both in vitro and in tumor models. Osteopontin (OPN), a mediator associated with tumor progression in different tumor types, was among the top upregulated genes in both cells and tumors and was the most overexpressed gene coding for a secreted protein. We further show that Slug is a direct regulator of osteopontin via binding to the OPN promoter. Interestingly, Slug expression and osteopontin secretion were correlated in vitro, as well as in tumor models, suggesting that liquid biopsies may be useful in estimating the aggressiveness phenotype of the tumor.

**Abstract:**

In colorectal cancer (CRC), disease-related death is closely linked to tumor aggressiveness and metastasis. Gene expression profiling of patient tumors has suggested that a more mesenchymal phenotype, present in about one-fourth of all patients, is associated with increased aggressiveness. Accordingly, the mesenchymal transcription factor Slug/*SNAI2* has been associated with decreased disease-free survival. To decipher the basis for the Slug-mediated phenotype, we conducted RNAseq experiments with a panel of HT-29 CRC cells expressing different levels of Slug, both in vitro and in tumor models. The results show that osteopontin, a secreted pleotropic protein involved in multiple steps of colorectal cancer progression, was highly upregulated by Slug in vitro, as well as in vivo. We further show that Slug is a direct regulator of osteopontin at the promoter level. The levels of secreted osteopontin were correlated with Slug expression, thereby linking the tumor phenotype to a biomarker available by liquid biopsies. The results also suggest that osteopontin neutralization may attenuate at least some of the Slug-mediated functions.

## 1. Introduction

Disease-related death in patients with colorectal cancer (CRC) is principally linked to metastasis [1], a multifactorial process during which cells from the primary tumor disseminate through the blood to distant sites, usually the liver and, less often, to the lungs and the peritoneum [2]. Whereas the 5-year survival rate is more than 90% for CRC patients during early disease stages, the survival rate drops to 7% for patients with metastatic disease (mCRC), corresponding to about half of all patients [1]. Despite important progress in the management of mCRC patients with respect to chemotherapeutic regimens and targeted therapies [3,4], there is still an urgent medical need for further advances to improve the outcome for this patient group. One important mediator of invasion and metastasis is the epithelial–mesenchymal transition (EMT), which combines the downregulation of epithelial proteins involved in cell adherence and the acquisition of mesenchymal properties such as the capacity of cells to migrate and disseminate. EMT is a key physiological process during early embryonic metazoan development, orchestrated by a number of different transcription factors including Snail/SNAI1, Slug/SNAI2, zinc-finger E-Box homeobox 1 and 2 (ZEB1 and ZEB2), and Twist-related protein 1 TWIST1 [5,6]. These factors are aberrantly reactivated in many types of cancer and have been associated with cancer progression and metastasis [5]. An algorithm was developed for EMT scoring based on the transcriptome signature for a wide variety of CRC tumor cell lines and patient tumors [7]. These studies identified EMT as an indicator of poor disease-free survival. Interestingly, a different study based on transcriptome analysis of a large collection of CRC patient tumors identified a mesenchymal subgroup with poor disease-free survival [8]. This group was characterized by 237 upregulated genes, including two mesenchymal transcription factors, Slug/SNAI2 and ZEB1. Consultation of the TCGA database (portal.gdc.gov (accessed on 15 April 2022) and the Human Protein Atlas (proteinatlas.org (accessed on 15 April 2022) for protein expression reveals that Slug, but not ZEB1, is expressed at the protein level in CRC. Slug/SNAI2 is a 30 kDa protein, initially identified in chick embryos and other cells undergoing EMT during development [9]. Slug is a member of the Snail family of transcriptional regulators and contains five zinc finger domains that exhibit conserved structure in all vertebrates and recognize a canonical *cis* sequence called the E-box: 5′-CACC/GGTG-3′ [8]. The pathological activation of Slug has been associated with an invasive and metastatic phenotype in epithelial cancers. In particular, Slug has been associated with enhanced metastasis, high tumor grade, and relapse [10,11]. Furthermore, some studies suggest that Slug may be an independent negative factor in CRC [12].

To better understand the mechanistic basis for the Slug-mediated malignant phenotype, we used a genetic CRC cell panel expressing different levels of Slug. A global transcription profiling strategy (RNAseq) was used to determine the influence of Slug on gene expression. The results identify a secreted mediator, osteopontin, as one of the most upregulated genes both in vitro and in tumor samples. The OPN gene encodes a glycoprotein of the extracellular matrix that has been associated with migration and invasion in different cancer types including CRC [13,14,15]. In this study, we elucidate the mechanistic basis for the Slug-mediated upregulation of OPN and demonstrate a close association between Slug expression in the tumor and the levels of secreted osteopontin.

## 2. Materials and Methods

### 2.1. Plasmids

For the genetic Slug model, HT-29 CRC cells were transfected using Lipofectamine and Plus reagent, Invitrogen. Y. Tony Ip, University of Massachusetts Medical School, Worcester, MA, USA generously provided the pCDNA3.1-SLUG plasmid encoding human Slug and the control plasmid. PGL3-OPN (-213)-luc and PGL3-OPN (-1206)-luc containing the human proximal and distal OPN promoter, respectively, were a kind gift from Gerhart Ryffel and were provided by Addgene, Waterdown, MA, USA (plasmids # 31106 or 31107).

### 2.2. Cells

HT-29 cells were provided by Richard Camalier (National Cancer Institute, Bethesda, MD, USA). LS174T and LoVo were a kind gift from Richard Hamelin (Saint-Antoine Research Center, Paris, France), while SW-480 colon carcinoma were purchased from ATCC (American Type Culture Collections). HT-29 Slug1 or Slug2 and the HT-29 transfection controls were established by stable transfection of HT-29 cells with Slug or with empty vector, respectively. The cells were maintained in DMEM (Dulbecco modified Eagle’s medium) supplemented with 5% (*v*/*v*) FBS (fetal bovine serum) and 1% penicillin/streptomycin (InVitrogen Corporation, Waltham, MA, USA). All cells were grown at 37 °C in a humidified 5% CO_2_ incubator and replaced after 2 months in culture. Upon defrosting, cells were routinely tested for mycoplasma contamination by a mycoplasma detection kit (Lonza, Basel, Switzerland).

### 2.3. Tumor Xenogafts

Two million cells were injected into the right flank of athymic 6 weeks old female NMRI-Fox 1nu mice (Taconic, Rensselaer, NY, USA). A total of 24 animals were used in this work, divided into 4 groups (WT, Control, Slug1, Slug2). The mice were followed for 24 days from the injection of the cells. Animals were weighed, and the tumor size was determined three times per week as previously described [16]. Mice were sacrificed before the tumor volume reached 3000 mm^3^. All animals were used for the RNA expression experiment by qPCR, while three representative mice from each group were used for RNAseq, Western blotting, and ELISA experiments. Animals were treated according to institutional guidelines. All further information relative to animal ethics is provided in the “Institutional Review Board Statement” section at the end of this manuscript.

### 2.4. Viability Assay

Cellular viability was determined by the MTT assay (3-(4, 5-dimethyl-thiazol-2yl)-2, 5-diphenyltetrazolium bromide) as described previously [17]. Cells were seeded for 24 hrs, then treated for 120 h with different doses of recombinant osteopontin (OPN) from PreproTech, Cranbury, NJ, USA (0, 50, 100, 200 ng/mL) in 24-well plates in culture media containing 5% FBS at a density of 7000 cells per well. Cellular viability was determined by exposing the cells to the MTT tetrazolium salt for 3 h at 37 °C, and the formation of formazan was measured at 570 nm by a microplate reader (Tecan, Männedorf, Switzerland). All values are averages of at least 3 independent experiments, each done in triplicate.

### 2.5. SiRNA

Cells were seeded in a 12-well plate and transfected with a specific siSlug or with AllStars negative control siRNA (Qiagen, Hilden, Germany) according to the manufacturer. Briefly, cells were transfected with 41 nmol of siRNA. After 48 h, cells were harvested, and the expression of Slug and osteopontin were determined by Western blot analysis. Alternatively, cells were seeded in transwell plates for functional analysis.

### 2.6. Promoter Reporter Assay

The human OPN promoter sequence was obtained from the NCBI gene database. For luciferase reporter gene assays, cells were seeded in 12-well plates and transfected with plasmids using the Fugene HD reagent (Promega, Madisson, WI, USA). After 24 h, luciferase activities were measured using a luciferase assay (Promega, Madisson, WI, USA) and normalized for transfection efficiency by a β-galactosidase-expressing vector and the Galacto-Star system (ThermoFisher Scientific, Waltham, MA, USA).

### 2.7. Chromatin Immunoprecipitation, ChIP Assay

HT-29 control or HT-29 Slug1 cells were seeded at 5 × 10^6^ cells in a 100 mm dish. After 24 h, cells were fixed with 1% formaldehyde. Cells were then lysed in SDS buffer supplemented with the proteases inhibitors aprotinin and pepstatin A according to the manufacturer’s instructions (Millipore, Burlington, MA, USA). Lysates were sonicated (3 times for 10 s at 30 power) to shear the DNA into pieces of 200–1000 base pairs. The sonicates were pre-cleared in ChIP buffer containing protease inhibitors. The DNA–protein complexes were then incubated in the presence or absence of 1 µg anti-Slug antibody (sc X166476, Santa Cruz Biotechnology Inc., Dallas, TX, USA) and then with protein G/DNA salmon sperm beads, followed by elution and reverse cross-linking at 65 °C. DNA was extracted using the phenol–chloroform method. The detection of Slug-associated DNA was detected by qPCR. All primers used are indicated in Appendix A.

### 2.8. Real-time RT (Reverse Transcription)-PCR

Total RNA was extracted from cell or tumors using the TRIzol RNA purification reagent (Invitrogen, Waltham, MA, USA). The purification of RNA from the tumors was performed using TissueLyser II (Qiagen, Hilden, Germany). Briefly, the samples were placed on dry ice, and two TissueLyser beads (3 mm, Qiagen, Hilden, Germany) and 1 mL of TRIzol were added. The samples were then shaken twice for 2 min at 30 Hz.

RNA quantity and purity were determined using a spectrophotometer (DeNovix Inc., Wilmington, DE, USA). Total RNA (1 μg) from each sample was reverse-transcribed using hexanucleotides. Real-time RT–PCR measurements were performed with an Mx3000P apparatus (Agilent Technologies, Santa Clara, CA, USA) with the corresponding SYBR Green kit. PCR primers were designed with the Primer Blast or Primer BD programs and obtained from Promega, Madisson, WI, USA. Gene expression was normalized to β-actin. All primers used are indicated in Appendix A.

### 2.9. Western Blot

Cell extracts were obtained by lysis with RIPA buffer (0.5% sodium deoxycholate, 50 mM Tris-HCl; pH 8, 150 mM NaCl, 1% NP40, 0.1% SDS) supplemented with a protease inhibitor cocktail (Roche). Equal amounts of protein (10–50 μg/lane) were loaded into SDS-PAGE gels. After transfer onto nitrocellulose membrane, blots were incubated overnight at 4 °C with the following antibodies: anti-OPN antibody (ab8448 Abcam, Cambridge, UK), anti-β-catenin antibody (1/1000, Santa Cruz Biotechnology, Inc., Dallas, TX, USA), anti-SNAI2 (Novus Biologicals, Englewood, CO, USA) anti-actin-HRP antibody (1/2000, Santa Cruz Biotechnology, Inc., Dallas, TX, USA), and anti-CDH1 antibody (1/1000, Santa Cruz, Inc., Dallas, TX, USA), followed by incubation with a horseradish peroxidase-conjugated secondary antibody (goat anti-rabbit or anti-mouse at 1/2000 dilution, Cell Signaling, Danvers, MA, USA). Results were revealed with a chemiluminescence ECL detection system (Bio-Rad, Hercules, CA, USA) and visualized on Chemidoc systems (Bio-Rad, Hercules, CA, USA). Protein expression was quantified by densitometric analysis of the immunoblots using Image Lab software v.5.2.1 developed by Bio-Rad, Hercules, CA, USA.

### 2.10. ELISA

Secreted human OPN from cells and tumor xenografts were measured using sandwich ELISA according to the manufacturer’s instructions (Bio-Techne, Minneapolis, MI, USA). OPN detection from cells was carried out as follows. One million cells were seeded on a Petri dish in serum-free media, incubated for 24 h, and the conditioned media was harvested. For tumor samples, total protein from the tumors were extracted by lysis with MPER buffer (25 mM bicine pH 7.6) (ThermoFisher Scientific, Waltham, MA, USA) supplemented with protease inhibitor cocktail (Roche) in 2 mL tubes, and two beads (tissueLyser beads 3 mm, Qiagen, Hilden, Germany)) were added. The lysis was carried out using TissueLyser II (Qiagen, Hilden, Germany)) twice for 2 min at 30 Hz. Lysates were then centrifuged for 15 min at 12,000 rpm at 4 °C, and the supernatants were used for ELISA. The circulating OPN was measured after the collection of serum from tumor-bearing mice.

For quantification of human OPN, ELISA was performed using isoform-specific capture and detection antibodies according to the manufacturer’s instructions. The results were normalized to the quantity of total protein except for serum OPN, for which the results are indicated as pg/mL serum.

### 2.11. Immunostaining

For immunocytochemistry, cells were seeded on coverslips for 24 h. For immunohistochemistry, paraffin-embedded tissue blocks were cut into 4 μm sections and transferred to glass slides. The slides were deparaffinized and rehydrated by incubation with three incubations with xylene (10 min each); three incubations with ethanol at 100%, 95%, 85%, and 75% at 5 min each; and two incubations with distilled water. For antigen retrieval, sections were heated at 121 °C in citrate buffer. After washing with PBS, cells or tissue were fixed with 4% (*v*/*v*) paraformaldehyde in PBS for 20 min, washed twice with PBS-0.1% Tween 20, and permeabilized with 0.5% Triton ×100 for 15 min at room temperature. Then, slides were washed and blocked for 30 min in 0.5% bovine serum albumin in PBS and incubated with anti-OPN or anti-Slug antibodies overnight. After three washes with PBS-0.5% Tween 20, tissue sections or cells were incubated with Cy3-conjugated secondary antibodies or with alexa Fluo-FITC (Jackson ImmunoResearch laboratories, West Grove, PA, USA) and then counterstained with DAPI at 1 μg/mL (4′,6′-diamidino-2-phenylindole), mounted with Vectashield (Vector Laboratories, Burlingame, CA, USA) for the cells or with glycerol for the tissues and observed by microscopy. Fluorescent images were captured using a fluorescence microscope Bx61 (Olympus, Tokyo, Japan).

### 2.12. Migration and Invasion

*Migration:* Cells were seeded into 24 wells at a density 2 × 10^5^ cells/well with serum-free media in the upper chamber (0.8 μm pore size, Corning, Corning, NY, USA) and media containing 10% FBS in the lower chamber. *Invasion:* Cells were plated in serum-free medium on transwell inserts (Corning, NY, USA) coated with 5 μg of matrigel.

After 24 h incubation, cells that had migrated or invaded into the lower surface of the insert were fixed and stained using Kwik-Diff kit (Shandon, Thermo Scientific, Waltham, MA, USA). The number of migrating/invasive cells was counted in five representatives fields per insert at ×100 magnification.

### 2.13. RNA-Sequencing and Pathway Enrichment Analysis

RNAseq experiments were done on iGenSeq plaform (genotyping and sequencing core facility) of the Brain and Spine Institute, Pitié-Salpêtrière, Paris, France. Total RNA was extracted using a mirVana miRNA Isolation Kit from Ambion, ThermoFisher Scientific, Waltham, MA, USA. Library preparation was performed using KAPA mRNA HyperPrep kit, Roche. Quality control was made using Tapestation, Agilent Technologies, and dosed using the Quantus/Quantifluor system from Promega, Madisson, WI, USA. Then, sequencing was carried out using the NextSeq 500 with a High Output kit v2 on 400 billion read-50 Gb array (Illumina Inc., San Diego, CA, USA). Paired-end reads were aligned to hg38 genome using STAR v2.5.3a [18] and transcripts quantification using RSEM v1.3 [19] based on Gencode v30 annotation. Enrichment analysis was performed using Enrich [20].

### 2.14. Statistical Analysis

Each experiment was repeated at least three times independently. All statistical analyses were performed with Prism software v.6.0 (GraphPad Software, San Diego, CA, USA). Averaged data were reported as means ± SEM. For comparisons between two groups, a one-way ANOVA test was used, and a multiple *t*-test was used to compare columns. Statistical significance was accepted for *p* < 0.05. Symbols: * *p* < 0.05; ** *p* < 0.01; *** *p* < 0.001.

## 3. Results

### 3.1. Global Gene Expression Analysis in Slug/SNAI2-Transfected HT-29 Cells

To establish the role of Slug in CRC, we created a genetic CRC model for Slug expression. As parental cells, we selected HT-29 cells that display a pronounced epithelial phenotype. HT-29 cells were transfected with Slug, and two stable Slug-expressing clones (Slug1, Slug2) were isolated (Figure 1A,B). As transfection control, we used HT-29 cells transfected with empty vector (control). To characterize the influence of Slug on gene expression, transcriptome analysis was performed for the four HT-29 cell lines as well as for the corresponding tumor xenografts. Global expression profiling showed that Slug-overexpressing cells and tumors were clustered together while the parental and control cells and tumors formed a separate cluster (Figure 1C). Next, a two-step analysis was carried out. First, a subtractive analysis was carried out comparing the gene profiles of Slug-expressing cells and tumors with the corresponding parental/control samples. Genes were considered to be significantly upregulated by Slug if the fold change (FC) was greater than +2 and downregulated if the fold change was less than −2 with *p*-values (false discovery rates) less than 0.01 (Figure 1D). For the first subset (cell lines), we found that 218 genes were upregulated by Slug, while 346 genes were downregulated (Appendix A). For the second subset (tumor xenografts), we found a significantly higher number of Slug-regulated genes, including 1234 upregulated genes and 886 downregulated genes (Appendix A). In a second step, we extended the analysis by comparing the two sets of results with each other. This strategy allowed us to focalize on the genes that were common to the two subsets. The results revealed that Slug expression led to the upregulation of 109 genes (including 104 annotated by DAVID and 84 encoding proteins, see Appendix A) and the downregulation of 110 genes in common for cells and tumors (Figure 1E). Importantly, we observed that Slug was among the most upregulated genes in the Slug transfectants with an FC of +27.67 for the cellular subset and an FC of +12.33 for the xenograft subset (Appendix A), thereby confirming the biological relevance of our Slug model and its characterization.

The upregulated genes were enriched in five different signaling pathways, including (1) the cAMP-signaling pathway, (2) focal adhesion, (3) extracellular matrix (ECM)–receptor interaction, (4) the estrogen-signaling pathway, and (5) the PI3K/Akt-signaling pathway (Figure 1F). Interestingly, we found that the *SPP1* gene, which codes for osteopontin (OPN), was third, with an FC of +33.69 for the first subset (cells), and first, with an FC of +44.74 for the second subset (tumor xenografts). SPP1/OPN contributes to three of the five signaling pathways identified above, including focal adhesion, ECM–receptor interaction, and the PI3K-AKT signaling pathway.

Next, a meta-cohort including the gene expression of 1820 CRC patient tumors [7] was used to determine if the expression of OPN/SPP1 was correlated with clinical parameters. The results indicate that OPN expression is weakly correlated with overall survival, with a hazard ratio of 1.6 and a *p*-value of 0.0022. In contrast, the difference is more marked for disease-free survival (DFS), with an HR of 1.72 and a *p*-value of 0.0008 (Figure 1G).

### 3.2. Influence of Slug/SNAI2 on OPN Expression in CRC Cells with Different Genetic Background

To characterize the correlation between Slug and OPN expression in CRC cells with different genetic backgrounds, we selected a minipanel of CRC cells. This includes HT-29 and LoVo cells that have low levels of Slug expression, as well as limited migratory and invasive capacities, and SW480 and LS174T cells that express more Slug and show a higher migratory and invasive potential. The basal expression of Slug and OPN were determined by RT-qPCR for each cell line (Figure 2A). The endogenous expression of *SLUG* and *OPN* were comparable with a higher expression for LS174T and SW480 cells (4–60-fold higher, respectively) compared to HT-29 and LoVo cells (Figure 2A). In agreement, the protein expression of both Slug and OPN were 3- to 4-fold higher in LS174T and SW480 cells compared to HT-29 and LoVo cells (Figure 2B).

To determine if Slug is able to regulate OPN, we carried out a gain-of-function approach using transient transfections with a Slug-expression vector. Slug induction was accompanied by increased OPN expression in all cell lines, which was most marked for SW480 and LS174T cells (Figure 2C). These findings were further confirmed by a loss-of-function approach using small interfering RNA targeting Slug in the two cell lines with the highest level of Slug expression, namely SW480 or LS174T. The downregulation of Slug was accompanied by a 50% decrease of OPN protein in SW480 cells (Figure 2D, upper panels) and a 70% decrease in LS174T cells (Figure 2D, lower panels). These results strongly suggest that OPN is, at least in part, regulated by Slug in CRC cells independent of the genetic background.

### 3.3. Slug/SNAI2 Increases the Expression and Secretion of Osteopontin

To further characterize the influence of Slug on OPN expression, we examined the expression of the two proteins in single cells using our genetic HT-29 model (Figure 2). The results show that Slug transfection is accompanied by the increased expression of OPN mRNA (Figure 3A) and protein (Figure 3B) for both Slug1 and Slug2 cells. Next, immunocytochemistry was used to determine the subcellular localization of the two proteins. The results revealed the homogeneous expression of Slug (red) in Slug1 and Slug2 cells, whereas the signal was barely detectable for parental and control cells (Figure 3C). Slug expression was accompanied by positive labeling for OPN (green) in the majority of Slug-expressing cells. At higher (100×) magnification, we observed a bright punctiform labeling of OPN (Figure 3D), suggesting that OPN might be contained in secretory vesicles For further confirmation, the levels of OPN secreted into the growth media were determined by ELISA analysis. The results revealed that increased Slug expression was accompanied by an important increase in OPN secretion, for both Slug1 and Slug2 cells, compared to the parental and control cells (Figure 3E).

### 3.4. Influence of Slug/SNAI2 on the Expression of OPN Isoforms, SIBLING Genes, and other Bone-Related Genes

OPN can exist as five different isoforms, *OPN-a*, *OPN-b*, *OPN-c*, isoform 4, and isoform 5 (Figure 4A), and some isoforms have been reported to be cancer-associated [18]. Isoform analysis revealed that Slug increased the expression of the full-length protein (*OPN-a*) and, to a lesser extent, the truncated *OPN-b* isoform. In comparison, the expression of the other variants was either low or undetectable (Figure 4B).

As the name indicates, osteopontin is a bone-related gene that is clustered with other bone-related genes coding for the SIBLING (small integrin-binding ligand N-linked glycoprotein) protein family (Figure 4C). This includes bone sialoprotein (*BSP*), dentin matrix protein 1 (*DMP1*), dentin sialoprotein (*DSPP*), matrix extracellular phosphoglycoprotein (*MEPE*), enamelin (*ENAM*), and Osteopontin (*OPN*). It is generally believed that, at least in bone, this gene cluster is co-regulated. However, in spite of the Slug-mediated upregulation of OPN, Slug had no detectable influence on the expression levels of the other SIBLING genes (Figure 4D). We then determined if Slug is able to regulate other genes involved in extracellular matrix formation and bone development. The genes tested include the transmembrane glycoprotein NMB (*GPNMB),* also known as osteoactivin; the bone gamma-carboxyglutamate protein (*BGLAP),* also known as osteocalcin; the two main bone-related collagen genes (*COL1A1* and *COL1A2)*; the secreted protein acidic and cysteine rich (*SPARC),* also known as osteonectin; and tenascin-C (*TNC),* an extracellular matrix glycoprotein. Generally, Slug had no detectable influence on the expression of any of these genes with exception of tenascin-C, which was 2- to 3-fold upregulated in Slug1 and Slug2 cells (Figure 4E). Taken together, these data suggest that Slug specifically regulates OPN expression in CRC cells independently of the osteogenic program or the SIBLING gene cluster.

### 3.5. Slug/SNAI2 Directly Binds to the OPN Promoter and Enhances Its Transcriptional Activity

Next, we wished to establish if Slug is able to directly regulate *OPN* expression at the promoter level. We used two luciferase gene reporters under the control of a short proximal (−213/+87) or a longer (−1206/+87) OPN promoter. An empty luciferase reporter was used as negative control. The results show that Slug expression is accompanied by increased transcriptional activity of the OPN promoter, which is most pronounced for the longer promoter (Figure 5A). Analysis of the OPN promoter sequence revealed the presence of three potential e-boxes between −150 and −50 bp (Figure 5B).

Next, ChIP-qPCR analysis was carried out to establish if Slug is able to bind directly to the *OPN* promoter. The results show that the addition of an anti-Slug antibody leads to the coprecipatation of the OPN promoter with Slug in Slug-expressing cells but not in parental or control cells (Figure 5C). These findings clearly indicate that Slug is able to bind directly to the OPN promoter.

### 3.6. The Influence of Osteopontin on Cellular Growth, Migration and Invasion

Next, we wished to establish how the secreted osteopontin can influence the biological functions of CRC cells. For this we selected two CRC cell lines, one (HT-29) displaying an epithelial phenotype and the other (SW480) with a more mesenchymal phenotype. The results show that the presence of biologically relevant concentrations of recombinant osteopontin had no detectable influence on the proliferation of either HT-29 (Figure 6A) or SW480 cells (Figure 6B).

Next, the trans-well assay (Boyden chamber) was used to determine the influence of osteopontin on cellular migration. Under standard conditions (with fetal calf serum in the lower well as chemoattractant), the presence of osteopontin enhanced cellular migration in a dose-dependent manner for both HT-29 (Figure 6C) and SW480 cells (Figure 6D). Unexpectedly, osteopontin was also able to stimulate migration in the absence of serum (Figure 6E,F).

The influence of osteopontin on invasion was determined using a modified Boyden chamber where the porous filter was overlaid with a thin layer of extracellular matrix before seeding the cells into the top chamber. The results show that osteopontin was able to stimulate the invasion of HT-29 and SW480 cells in a dose-dependent manner both in the presence (Figure 6G,H) and absence of serum (Figure 6I,J)

### 3.7. Slug/SNAI2 Upregulates OPN in Human Tumor Xenografts

We subsequently wanted to establish if Slug influences OPN expression in vivo. Human tumor xenografts (n = 19 for each tumor model) were established in nude mice using our genetic HT-29 slug model. Four weeks after injection, the expression of tumor-associated Slug and OPN was determined by RT-qPCR using human primers. The results show that Slug and OPN expression is comparable between parental and control tumors. In clear contrast, Slug was upregulated ~100-fold in xenografts from Slug1 and Slug2, while OPN was upregulated 100- to-200-fold (Figure 7A,B). Then, tumor extracts were prepared, and the expression of OPN protein was determined by ELISA analysis. The results (Figure 7C) are coherent with the mRNA data for both Slug1 and Slug2 tumors showing a significant upregulation of OPN protein. These findings were further confirmed by immunohistochemistry (Figure 7D). Specifically, the IHC results show comparable levels of OPN protein in tumors from parental and control cells (Figure 7D, upper panels, light brown staining), whereas osteopontin was highly expressed in Slug1 and Slug2 tumors (Figure 7D, intense brown staining).

To establish if tumor expression of OPN is reflected in the levels of circulating OPN, serum was collected from the tumor-bearing mice, and OPN was determined by ELISA. The results show a strong increase in circulating OPN in mice with Slug1 and Slug2 tumor xenografts compared to mice carrying tumors from parental or control cells (Figure 7E).

## 4. Discussion

We here report a mechanistic link between two factors associated with poor prognosis, invasion and metastasis, in patients with colorectal cancer. We show that the upregulation of the mesenchymal transcription factor Slug is accompanied by the increased expression of osteopontin in both tumor cell models and in human tumor xenografts. We subsequently demonstrate that Slug activates osteopontin via direct binding to cis regulatory sequences in the osteopontin promoter. Importantly, tumor expression of Slug was accompanied by increased osteopontin secretion and thus, in the levels of circulating osteopontin. We subsequently showed that osteopontin was a strong inducer of tumor cell migration and invasion in CRC cells with different phenotypes and genetic background. These results suggest that at least some of the Slug-associated tumor-promoting functions may be mediated by osteopontin.

Both Slug and osteopontin are associated with tumor progression in CRC [9,10,11], as well in as other tumor types, including breast cancer [19], hepatocarcinoma [20], and lung cancer [21]. Although it has been reported that Slug is able to induce some osteogenic genes like OPN, osteocalcin and collagen 1 via regulation of RUNX2 [22], a direct link between Slug and OPN has, to the best of our knowledge, never been described before. In this work, we used reporter plasmids and ChIP assays to show that Slug binds directly to the OPN promoter, thereby identifying a new Slug target gene. Isoform analysis indicated that Slug preferentially promoted the expression of OPN-a and OPN-b isoforms. It is currently believed that the functions of OPN variants depend on the specific cellular context, since only full-length OPN (OPN-a) display pro-inflammatory activities [23,24]. Interestingly, although osteopontin is part of the SIBLING gene cluster, Slug showed marginal, if any, influence on the expression of the other osteogenic genes in CRC cells.

Recently, it has been suggested that Slug may repress OPN in normal bone marrow [25]. The discrepancy with our findings underlines the importance of both the cellular context and the microenvironment, which are known to influence the transcriptional preferences of zinc-finger protein transcription factors. Specific E-box recognition and affinity is believed to depend on microenvironmental cues, with Slug being able to act either as an activator or a repressor of gene expression [25]. Interestingly, although Slug is mostly known as a gene repressor, in particular with respect to E-cadherin and other epithelial genes [26,27], several studies have demonstrated the capacity of Slug to activate gene transcription in a cell type- and microenvironment-specific manner. Genes subject to upregulation by Slug includes Vimentin [27], ZEB1 [28], CXCR4 [29], and PLD2 [30] as well as the SNAI2 gene itself [31]. The exact mechanism by which Slug serves as a transcriptional modulator remains elusive. However, it has been suggested that Slug is able to recruit other transcriptional regulators to the promoter regions. Furthermore, it has been suggested that Slug is also able recruit epigenetic regulators like the histone deacetylase HDAC1 or the lysine-specific demethylase LSD1 [27]. Further investigation is needed to better understand how Slug regulates the transcriptional program under different conditions.

In this work, we also show that osteopontin is a potent activator of tumor cell migration and invasion in CRC cells with different genetic background and different phenotypes (both epithelial and mesenchymal). These findings confirm and expand the findings of others [32]. The capacity of osteopontin to stimulate tumor cell invasion could explain at least some of the tumor-promoting activities of this mediator.

Characterization of human xenografts in nude mice revealed that the increased expression of Slug in the tumor is accompanied by increased levels of osteopontin in the tumor, as well as in circulation, thereby linking Slug expression to a circulating biomarker amenable to liquid biopsies. However, osteopontin may also be overexpressed during chronic inflammation, thereby limiting its use as a cancer-specific biomarker for the detection of CRC [13]. In spite of these limitations, our findings raise the intriguing possibility that at least osteopontin-neutralizing antibodies in patients with advanced CRC might neutralize some of the tumor-promoting effects of Slug.

Previous studies have shown that OPN expression is associated with lymph node metastasis, postoperative metastasis, and poor survival. A meta-analysis of more than 228 publications revealed that high OPN levels in plasma or tissue were correlated with decreased overall survival (OS) and disease-free survival (DFS) in different tumor types, including lung cancer, breast cancer, head and neck cancer, liver cancer, and prostate cancer [33]. Another meta-analysis, including 15 publications representing 1698 CRC samples, showed a strong correlation between the levels of OPN and tumor grade, invasiveness, DFS, and OS in CRC [15]. In apparent contrast, a study of 222 cases of well-characterized colorectal carcinomas (Stages I–III), evaluated by microdensitometry, showed a moderate survival advantage for patients with high expression of OPN [34]. However, as pointed out by the authors, due to the heterogeneity of epitope expression within the tumor, microdensitometry may not be a precise technique for the quantification of proteins, compared to immunohistochemistry. Since stage IV tumors were not included in this study, an alternative explanation of these findings could be that OPN (like TGFβ) might have biphasic effects, being tumor-suppressive in early stages and tumor-promoting later on.

## 5. Conclusions

The present study shows for the first time that OPN is regulated by the mesenchymal transcription factor Slug/SNAI2, thereby providing a direct link between two biomarkers associated with aggressive colorectal cancer. Slug directly upregulated osteopontin expression at the promoter level but had no, or only marginal, effect on other osteogenic genes in the SIBLING gene cluster. Increased osteopontin levels promoted migration and invasion in CRC cells independent of phenotype or genetic background. We finally show that the increased expression of Slug in CRC xenografts are accompanied by the increased expression of osteopontin both in the tumor and in circulation, thereby linking the expression of a transcription factor in the tumor to a biomarker potentially useful for liquid biopsies.

## Figures and Tables

**Figure 1 cells-11-01808-f001:**
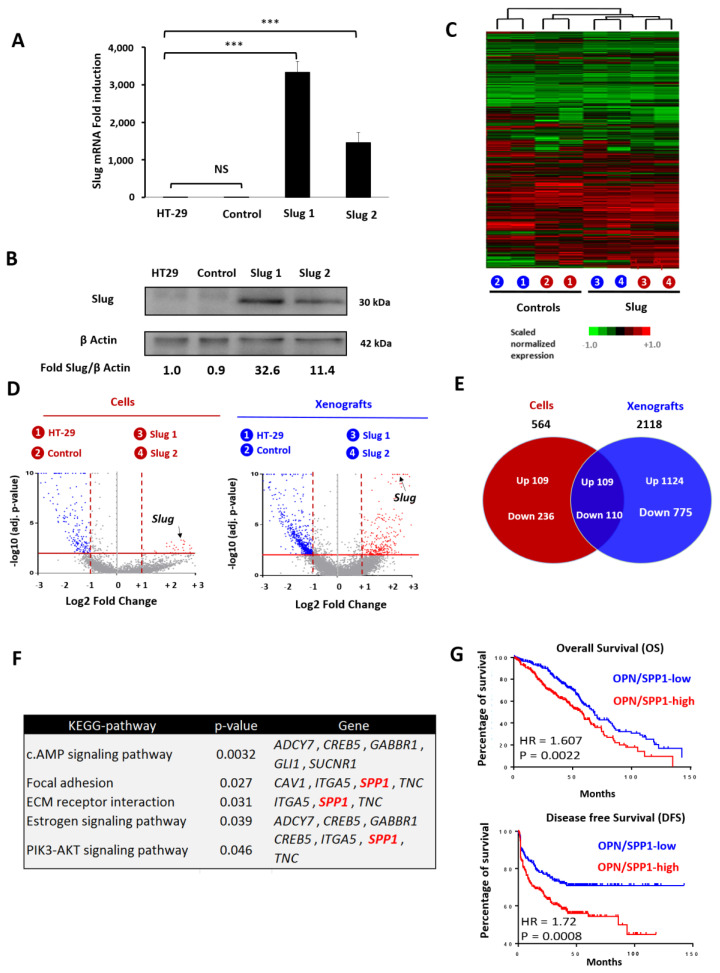
Influence of Slug/SNAI2 on global gene expression in CRC cells and tumors. (**A**) Expression of Slug mRNA in a genetic HT-29 model, including the parental HT-29 cells, the empty plasmid control and two stable Slug transfectants, Slug1 and Slug2. Symbols: *** *p* < 0.001 (**B**) Slug protein expression in the same four HT-29 models mentioned above. β-actin was used as loading control. (**C**) Heat map of the genes expressed by Slug1 and Slug2 cells and tumors compared to parental and control cells. Hierarchical clustering was performed by using the one minus Pearson correlation. Genes are considered to be significantly upregulated in the Slug transfectants, compared to controls, if the fold change (FC) is greater than 2, and considered to be downregulated if the FC is less than −2 with *p*-values (false discovery rates) less than 0.01. (**D**) Volcano plots showing the significant genes present in the indicated cells or tumors. Downregulated genes are depicted in blue, and upregulated genes in red, while Slug expression is indicated by a red arrow (log2 fold-change (FC) vs. –l-log10 *p*-value). (**E**) Venn diagrams of the number of up-or downregulated genes in cells, tumors, and in both groups. (**F**) KEGG-pathway enrichment analysis of the up-regulated genes. *SPP1* encoding osteopontin (OPN) is part of the signature for focal adhesion, extracellular matrix (ECM)–receptor interaction, and the PI3K-AKT signaling pathway. (**G**) Kaplan–Meier curves of overall survival (OS) on the left and disease-free survival (DFS) on the right, based on the expression of OPN in a meta-cohort of 1820 CRC patients [7].

**Figure 2 cells-11-01808-f002:**
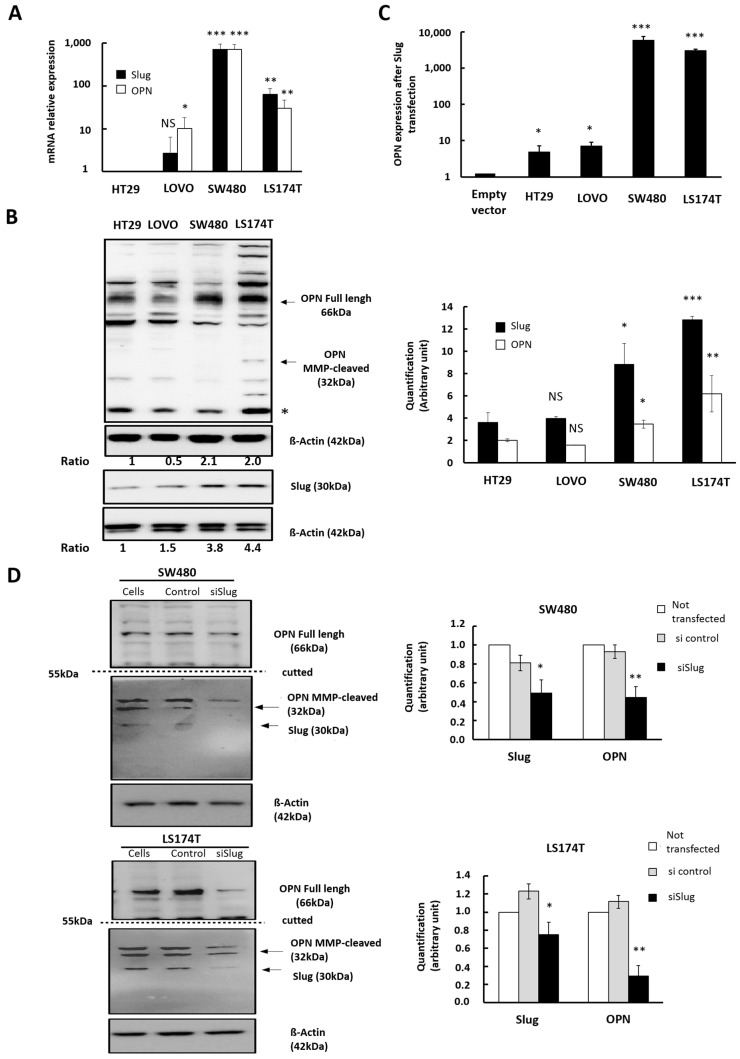
Influence of Slug/SNAI2 on OPN expression in CRC cells with different genetic background. (**A**). Relative expression of OPN mRNA following transitory transfection of Slug as compared to empty vector for the 4 CRC cell lines (HT-29, LoVo, SW480, and L174T). The graph represents the mean of at least 3 independent experiments, and the error-bars represent the standard error (SEM). Data were considered significant if p was less than 0.05 as determined by the ANOVA test. Symbols: * *p* < 0.05, ** *p* < 0.01, *** *p* < 0.001, NS, not significant. (**B**) Western blot analysis of Slug and osteopontin proteins in HT-29, LoVo, SW480, and LS174T cells. (*) Extra band: putative 15 kDa cleaved form of OPN, full length OPN, MMP cleaved form of OPN. β-actin was used as loading control. (**C**) Relative expression of OPN mRNA following transitory transfected Slug-encoding expression vector compared to empty vector in the 4 CRC cell lines (HT-29, LoVo, SW480, and L174T). The graph represents the mean of at least 3 independent experiments, and the error-bar represent the standard error (SEM). Data were considered significant if p was less than 0.05 as determined by the ANOVA test. Symbols: * *p* < 0.05, ** *p* < 0.01, *** *p* < 0.001, NS, not significant. (**D**) Western blot analysis of Slug and osteopontin protein 48 h after transfection of the indicated cells with si control or with si Slug. β-actin was used as loading control. The histogram indicates the mean of 2 independent experiments. The bar shows the SD. Symbols: * *p* < 0.05, ** *p* < 0.01, *** *p* < 0.001.

**Figure 3 cells-11-01808-f003:**
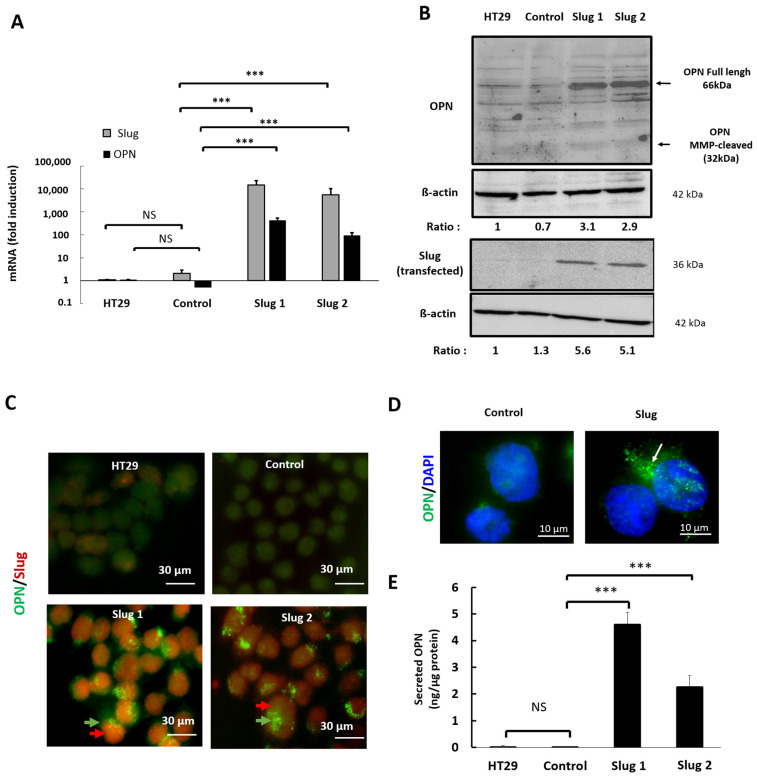
Slug/SNAI2 promotes the expression and secretion of OPN. HT-29 cells were stably transfected with a plasmid coding for Slug or with an empty expression vector (control). (**A**) The expression of Slug and OPN mRNA was determined by qRT-PCR, and the results were normalized to HT-29. (**B**) The expression of Slug and osteopontin protein was determined by Western blot analysis with β-actin as loading control. (**C**) Cells were stained with OPN-directed (green) and Slug-directed (red) antibodies, and the localization of the corresponding proteins was detected by immunofluorescence. Magnification ×100. Punctiform labeling of OPN is indicated with green arrows. (**D**) Cellular sub-localization of osteopontin in parental and Slug tranfectants as determined by immunofluorescence staining of osteopontin (green). Nuclei were stained with DAPI (blue). Punctiform labeling of OPN is indicated with white arrows. (**E**) Conditioned media were collected, and the amounts of secreted osteopontin were quantified by ELISA. Data were analyzed by the Student’s two-tailed *t*-test and considered significant when *p* was less than 0.05. Symbols: *** *p* < 0.001, NS, not significant.

**Figure 4 cells-11-01808-f004:**
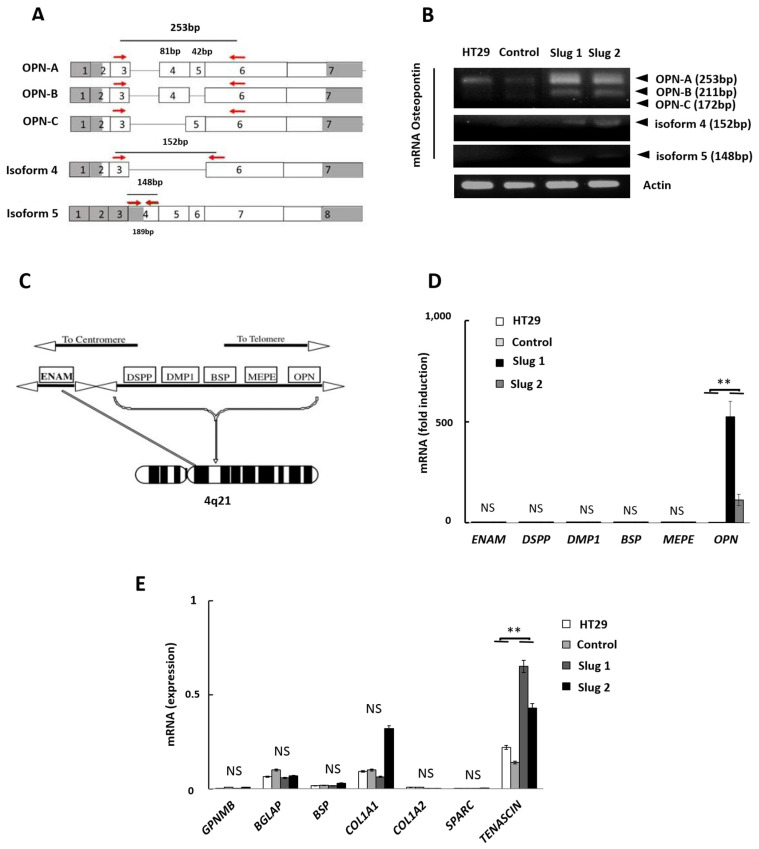
Slug/SNAI2 specifically increases the expression of OPN-A and OPN-B but no other SIBLING or osteogenic genes. (**A**) Diagram of the different OPN isoforms. The reading frames used as primers are indicated with red arrows. (**B**) Expression of OPN isoforms in the four HT-29 cell lines as determined by electrophoresis. Following qPCR, the different amplicons were loaded and separated on a 2% agarose-TAE gel. Actin was used for normalization. (**C**) The SIBLING family gene cluster: the order of the genes is as follows, going from the left (closest to the centromere) to the right (toward the telomere): ENAM, DSPP, DMP1, BSP, MEPE, and OPN. (**D**) qRT-PCR was performed to determine the influence of Slug on the expression of the SIBLING genes. The graph represents the mean of 3 independent experiments. The brackets indicate the difference between control and Slug1 or Slug2. Data were considered significant if p was less than 0.05, as determined by the ANOVA test. Symbols: ** *p* < 0.01, NS, not significant. (**E**) qRT-PCR analysis of additional osteogenic genes, including GPNMB BGLAP, BSP, COL1A1, SPARC, and TENASCIN C. The graph represents the mean of 3 independent experiments. The brackets indicate the difference between control and Slug1 or Slug2. Only the expression of tenascin C reached significance. Data were considered significant if *p* was less than 0.05, as determined by the ANOVA test. Symbols ** *p* < 0.01; NS, not significant.

**Figure 5 cells-11-01808-f005:**
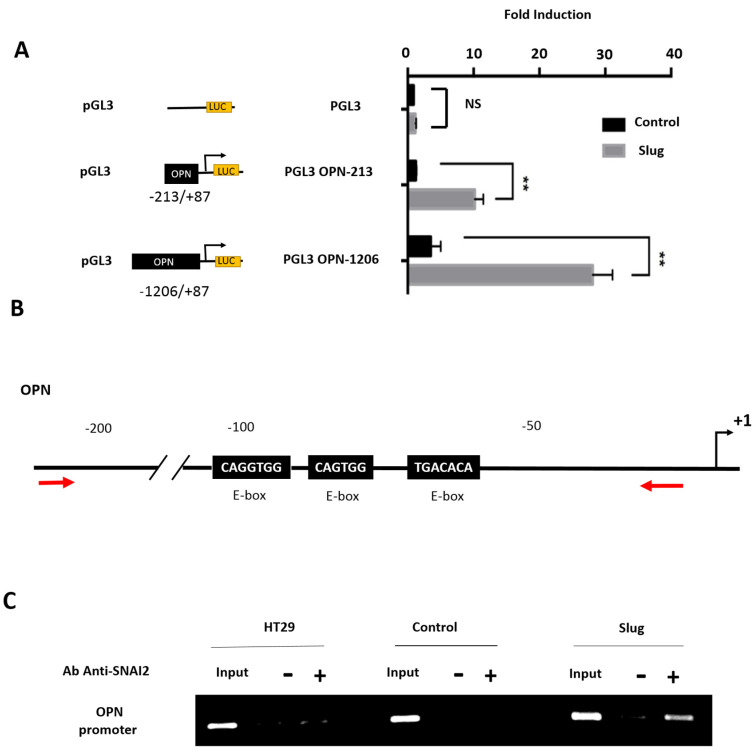
Slug/SNAI2 binds directly to the OPN promoter and enhances its transcriptional activity. (**A**) Luciferase activity in HT-29 control cells (black) and in Slug-transfected HT-29 cells (gray) of the proximal (−213/+87) and the distal (−1206/+87) OPN promoter. The fold induction was calculated after normalization with empty vector. Cells were co-transfected with RSV-β-galactosidase to normalize the luciferase activity. The graphs represent the mean of 3 independent experiments. Bars, SEM. Data were considered significant if p was less than 0.05 as determined by the two-tailed Student’s *t*-test. Symbols: ** *p* < 0.01, NS, not significant. (**B**) The proximal promoter region of OPN by BLAST, indicating the putative Slug binding sites (e-box) and the primers (red arrows) that were used. (**C**) Chromatin immunoprecipitation (ChIP) analysis with a Slug-directed antibody in parental, control, and Slug-transfected cells.

**Figure 6 cells-11-01808-f006:**
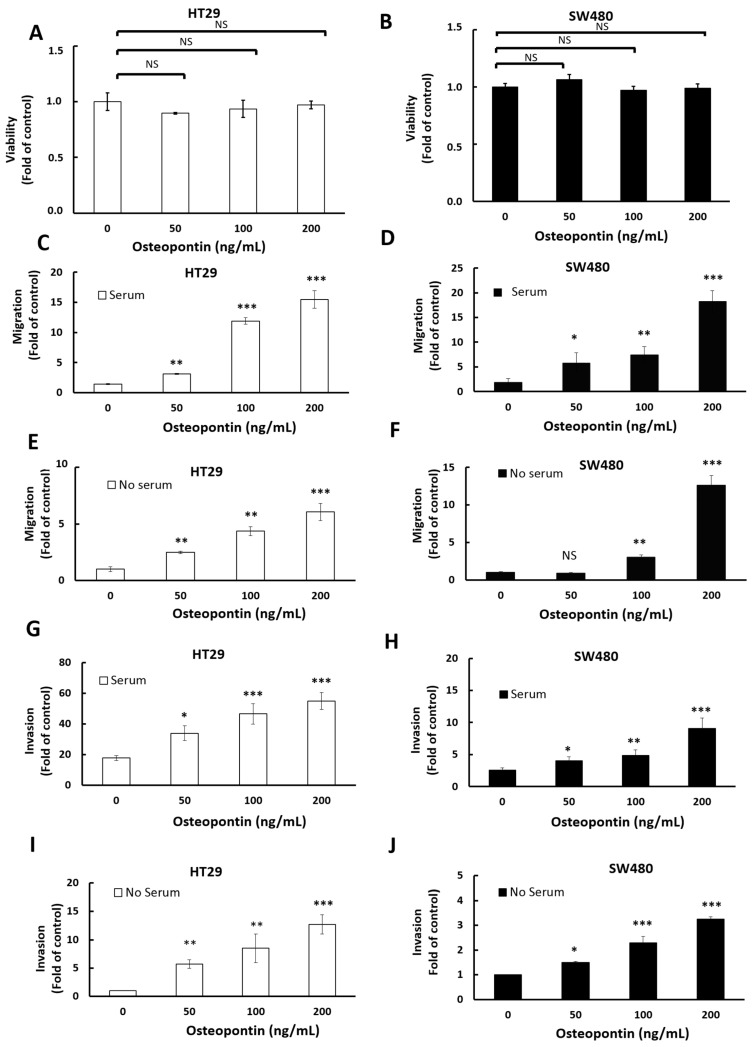
Osteopontin enhances the migration and invasion of HT-29 and SW480 cells but has no influence on cell growth. The influence of osteopontin on cellular growth, migration, and invasion was determined for the epithelial HT-29 cells (left, gray columns) and the more mesenchymal SW480 cells (right, black columns) (**A**,**B**) Cells were grown under standard conditions in the absence or presence of the indicated amounts of osteopontin for 96 h, followed by the MTT viability assay. All values are normalized to the HT-29 and SW480 control cells grown in the absence of osteopontin. (**C**,**D**) The migration of HT-29 and SW480 cells incubated with or without OPN (0, 50, 100, and 200 ng/mL) in the upper chamber and with serum in the lower chamber was determined after 24 h (SW480) or 48 h (HT-29). (**E**,**F**) The migration of HT-29 or SW480 cells incubated with or without OPN (0, 50, 100, and 200 ng/mL) in the upper chamber and without serum in the lower chamber was determined after 24 h (SW480) or 48 h (HT-29). The influence of osteopontin on cellular invasion was determined using a modified Boyden chamber where the porous filter is overlaid with a thin layer of extracellular matrix (matrigel) before seeding the cells into the top chamber. (**G**,**H**) The invasion of HT-29 and SW480 cells incubated with or without OPN (0, 50, 100 and 200 ng/mL) in the upper chamber and with serum in the lower chamber was measured after 24 h (SW480) or 48 h (HT-29). (**I**,**J**) The invasion of HT-29 and SW480 cells incubated with or without OPN (0, 50, 100, and 200 ng/mL) in the upper chamber and without serum in the lower chamber was measured after 24 h (SW480) or 48 h (HT-29). All graphs represent the mean of at least 3 independent experiments, each done in triplicate. Bars, SEM. Data were considered significant if *p* was less than 0.05 as determined by the two-tailed Student’s *t*-test. Symbols: * *p* < 0.05; ** *p* < 0.01; *** *p* < 0.001, NS: not significant.

**Figure 7 cells-11-01808-f007:**
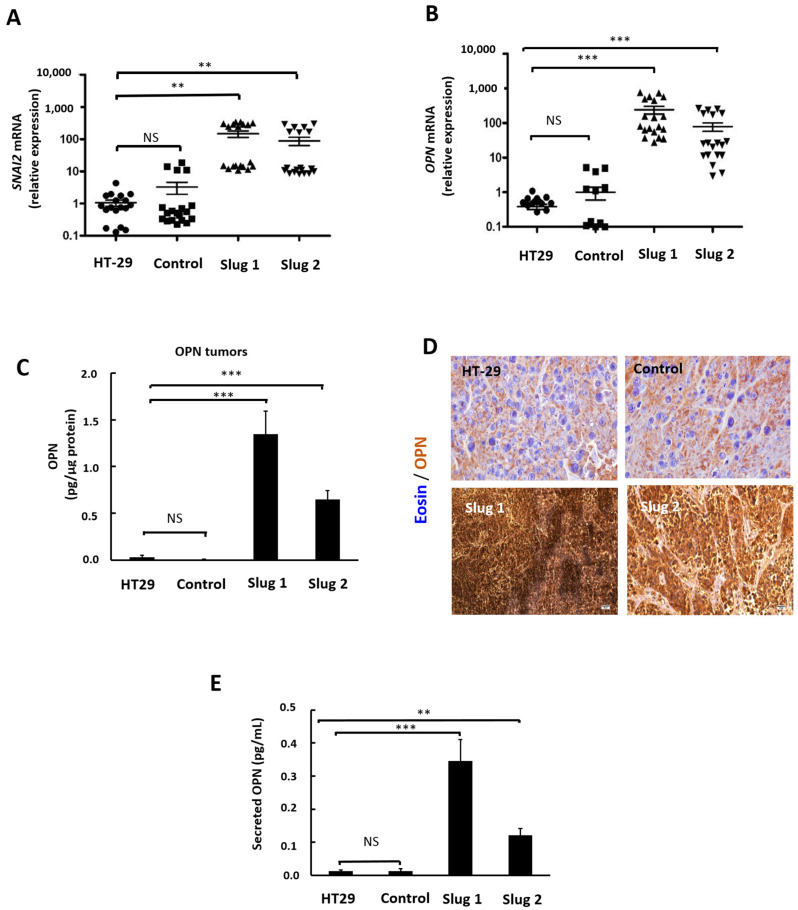
Influence of Slug on OPN expression in tumor xenografts. Tumor xenografts were established in nude mice from HT-29, control, Slug1, and Slug2 cells. Four weeks after injection, tumor xenografts and blood were collected for further analysis. (**A**,**B**) mRNA was extracted from the tumor xenografts, and the expression of mRNA for Slug (**A**) and OPN (**B**) was determined by qRT-PCR using human primers. The data represent a total of 76 different tumor samples. (**C**) The levels of tumor-associated OPN protein were determined by ELISA analysis. (**D**) Immunohistochemistry of OPN. A strong brown coloration indicates the presence of osteopontin. Nuclei were stained with eosin (blue). (**E**) Serum levels of OPN were quantified by ELISA analysis. The values represent the average of two independent experiments carried out for three mice in each groups in triplicate. Data were analyzed by the Student’s two-tailed *t*-test and considered significant when *p* was less than 0.05. Symbols: ** *p* < 0.01; *** *p* < 0.001, NS, not significant.

## Data Availability

RNAseq data are available in the ArrayExpress database http://www.ebi.ac.uk/arrayexpress under access number E-MTAB-11812.

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
