# Peer review of "Osteopontin (OPN/SPP1), a Mediator of Tumor Progression, Is Regulated by the Mesenchymal Transcription Factor Slug/SNAI2 in Colorectal Cancer (CRC)"

_cells, 2022, doi:10.3390/cells11111808_

Round 1

Reviewer 1 Report

The peer-reviewed paper describes the regulation of OPN by Slug in colon cancer. The authors used the HT-29 cell line overexpressing Slug to identify the genes regulated by Slug. Thus, they selected SPP1 (the gene encoding OPN) as potentially most exposed in Slug overexpressing cells. In further studies, they confirm the dependence of OPN on Slug.

I have some notes on the manuscript:

  1. The methodology poorly describes the study carried out on animals, did not provide the number of animals, the time of their observation, and the number/symbol of ethics committee approval for the study.
  2. It is also not described how the material from the tumors was prepared for the remaining studies.
  3. The results did not show the kinetics of tumor growth and body weight. This is ethically important. It is also not known whether the applied genetic modification of the HT-29 cell influenced the growth of tumors?
  4. Was the material for the global gene expression study (chapter 3.1 - unnamed) from the same mice described in chapter 3.1.7?
  5. The legends of Figs are chaotic. Especially Fig. 2 e.g. Fig 2C does not contain data for 2 vectors Slug1 and 2, which can be read in the legend.
  6. The authors in the text refer to Table 1 and Suppl. Table 1 and Suppl. Table 2. However, 4 tables are in the supplement. The authors need to sort this out.
  1. In addition, there are minor punctuation errors, etc. in the text.

Author Response

Dear colleague,

On behalf of authors, Dr. J. Denis.

Reviewer 2 Report

In the manuscript Amilca-Seba et al. identified a mechanistic link between two biomarkers associated with metastatic colon cancer: the mesenchymal transcription factor Slug/SNAI2, and the Osteopontin, a secreted pleotropic protein involved in multiple steps of colorectal cancer progression. The authors show, by reporter gene expression and chip assays, that Slug/SNAI2 is a direct regulator of Osteopontin at the promoter level. Furthermore, the levels of Osteopontin were correlated with Slug/SNAI2 expression both in CRC cell lines and xenografts in nude mice, as well the level of secreted Osteopontin.

The methods used in the study and the results are well described and discussed. Overall, the results provide a robust conclusion to strength what proposed by the Authors: that Osteopontin may be a valid biomarker available by liquid biopsies and that its neutralization may attenuate some of the Slug-mediated functions.

Minor suggestion:

  • The letters in figures 2, 6 and 7 are small and difficult to read, they could be enlarged.
  • Table 1 could be transferred in supplemental materials.

Author Response

(The authors gave the same response as above.)

Reviewer 3 Report

The work of Amilca-Seba et al. unravels a regulatory activity of slug on the expression of osteopontin. This is new and a little unexpected and thus an interesting contribution to the field.

However, since the overall discussion aims more towards clinical use of this observation, some major concerns apply. And the manuscript must be improved before publication:

Major points:

  • Point 3.1.2: the point that CRC cell lines are more or less aggressive in vivo is not really relevant. Add the data of cell doubling times of the SW480, LS174T, HT29 and LoVo lines used in your lab. Do the latter two divide slower than the first two? If yes, this would deliver a very simple alternative explanation for the differences found?
  • Concerning the prognostic value of osteopontin, we published in 2012 a paper in which we found: “With regard to prognosis, microdensitometric evaluation of a tissue microarray made of a clinicopathologically well-characterized series of colorectal carcinomas with long-term follow-up (222 cases evaluable in the tissue microarray, UICC Stages I-III/R0) showed a moderate survival advantage of patients with high osteopontin expression by microdensitometry. (Prall et al., 2012)”. Thus, at least it should be discussed, that not all studies come to the same conclusion as the authors state in their work concerning the prognostic value of high or low osteopontin expression.
  • Taking this into account, all arguments dealing with the conclusion: “… linking the tumor phenotype to a biomarker subject to liquid biopsies. The results also suggest that neutralization of osteopontin may attenuate at least some of the Slug-mediated tumor promoting functions.” is not really covered by data presented in the present work. It deals with unraveling a functional connection on the gene regulatory level between slug and osteopontin. The outlook towards clinical CRC management is not necessary at all and should simply be removed or at least substantially be reduced.

Minor points:

  • STR-typing data of cell lines used are missing.
  • Data of tumor xenografts are not really relevant since identical cells were used in vitro and in vivo. Superíor would for example have been data with true PDX model panels varying in slug expression levels.
  • “Point 3.1 subsection” is funny.

Author Response

(The authors gave the same response as above.)

Round 2

Reviewer 3 Report

My points have been properly addressed.

This manuscript is a resubmission of an earlier submission. The following is a list of the peer review reports and author responses from that submission.